# Effect of Processed Volcanic Ash as Active Mineral Addition for Cement Manufacture

**DOI:** 10.3390/ma15186305

**Published:** 2022-09-11

**Authors:** Julia Rosales, Manuel Rosales, José Luis Díaz-López, Francisco Agrela, Manuel Cabrera

**Affiliations:** Area of Construction Engineering, University of Cordoba, 14014 Cordoba, Spain

**Keywords:** volcanic ash, cement, pozzolanic behaviour, chemical composition, mechanical behaviour

## Abstract

In the last quarter of 2021, there was a very significant eruption of the Cumbre Vieja volcano on the island of La Palma, belonging to the Canary Islands, Spain. It generated a large amount of pyroclastic volcanic materials, which must be studied for their possible applicability. This work studies the properties and applicability of the lava and volcanic ash generated in this process. The need for reconstruction of the areas of the island that suffered from this environmental catastrophe is considered in this study from the point of view of the valuation of the waste generated. For this purpose, the possibility of using the fine fraction of ashes and lava as a supplementary cement material (SCM) in the manufacture of cement is investigated. The volcanic material showed a chemical composition and atomic structure suitable for replacing clinker in the manufacture of Portland cement. In this study, the cementing and pozzolanic reaction characteristics of unprocessed volcanic materials and those processed by crushing procedures are analysed. To evaluate the cementitious potential by analysing the mechanical behaviour, a comparison with other types of mineral additions (fly ash, silica fume, and limestone filler) commonly used in cement manufacture or previously studied was carried out. The results of this study show that volcanic materials are feasible to be used in the manufacture of cement, with up to a 22% increase in pozzolanicity from 28 to 90 days, showing the high potential as a long-term supplementary cementitious material in cement manufacturing, though it is necessary to carry out crushing processes that improve their pozzolanic behaviour.

## 1. Introduction

After 50 years of quiescence in La Palma Island (Canary Islands, Spain), the Cumbre Vieja volcano—historically the most active volcano in the Canary Islands—began an eruptive episode on 19 September 2021, forcing the evacuation of 7000 residents, destroying infrastructure worth more than EUR 400 m, and affecting 1.212 hectares and 92.7 km of roads with solidified lava and ash [1]. All this volcanic solidified lava and ash, together with growing environmental awareness and a circular economy, considering that the construction industry is perceived as a major contributor to environmental degradation [2] that consumes 40% of the raw materials extracted [3], makes the study of lava and ash for its application in building materials very interesting. These materials, formed from the cooling of magma from the volcanic eruption, are known as pyroclastic materials and have very heterogeneous physical properties, varying in particle size from microns (ash) to metres (solidified lava) [4], and can have a dense or vesicular structure [5,6].

Dingwell et al. [7] differentiated typical volcanic ashes as pyroclastic debris no larger than 2 mm, however, many authors carry out crushing and sieving procedures for the utilization of volcanic ash [8,9,10,11,12]. Lemougna et al. [10] ground volcanic ashes to pass a 400 μm sieve; Leonelli et al. [11] dry-milled the analysed volcanic ashes to a fineness of 150 mm; Tchakoute et al. [12] ground and sieved the ashes to a powder of 80 μm. Some authors have determined the influence of the particle size of volcanic ash for use as a construction binder. According to Moufti et al., [13], finely pulverised ash with a particle size of less than 45 mm and a content of 10% by mass has a compressive strength similar to a control sample. On the other hand, Khan et al. [14] reported that 15% substitution of natural pozzolans with finely ground cement had a lower strength compared to controls. An important property of this type of material is that it has pozzolanic activity, i.e., in contact with water it can behave as a hydraulic binder, just like cement [15,16]. 

Currently, the use of lava and volcanic ash has been evaluated by different authors for use as construction material; the main applications have been as ceramic material, geopolymers, cement, and concrete [17,18,19,20]. Zhang et al. [21], manufactured and analysed bricks fired with a mixture of volcanic ash and black cotton soil between 1000–1050 °C, showing good compressive strength (60 MPa), a small percentage of dimensional variation, and similar bulk density to conventional brick. Serra et al. [22] reported the use of ash as a flux for feldspar replacement in clay-based materials and observed appropriate brick texture and mechanical properties compared to traditional materials used in brick manufacturing. 

The high content of aluminosilicates for the synthesis of geopolymers has attracted the interest of a large number of studies of this type of mineral in the production of geopolymer materials, either as the sole source of aluminosilicate material [23,24] or combined with other types of materials such as metakaolin [25].

Furthermore, numerous studies corroborate the suitability of volcanic ash for partial replacement of cement, paste, and mortar or in the manufacture of concrete [25,26,27,28,29]. For example, Celik et al. [27] reported that a high-volume mass replacement of Portland cement (OPC) with volcanic ash produces concrete with good workability, high compressive strength, and high resistance to chloride penetration. Al-Fadala et al. [27] analysed the mixture of volcanic ash and cement according to international standards, to evaluate the use of this material, and concluded that it met the technical requirements to be used for certain percentages of volcanic ash from a chemical, physical, and mechanical point of view. Regarding treatments applied to volcanic ash prior to its use, Khan et al. [28] showed that pozzolanic activity increased with the fineness of the material; however, a heat treatment applied to volcanic ash was not positive. Other studies, such as that of Abdullah et al. [30], showed that volcanic pumice powder improved the compressive strength of self-compacting concretes made with it, thus demonstrating the influence of the degree of fineness of volcanic ashes on the mechanical properties. Al-Swaidani and Aliyan [31] studied the durability of mortar and concrete made with different slag substitutions, showing great interest in properties related to chloride ion penetration, acid attacks, and corrosion of reinforcing steel, and concluded that the volcanic slag studied was suitable for use as a natural pozzolan in accordance with international standards.

Therefore, taking into account that cement, and especially the process necessary to produce it, contributes significantly to climate change, emitting 8% of total CO_2_ emissions worldwide, the aim of this work is to study the use of ash from the Cumbre Vieja volcano as a replacement for cement in the production of Portland cement as well as its effects on the manufacture of mortar. The physical, chemical, mechanical, and environmental properties, in accordance with international specifications, have been studied. This study shows the long-term pozzolanic potential of volcanic ashes and how the application of a crushing treatment influences the mechanical properties of cement mortars. A comparative study has been carried out with other types of commonly used mineral additions. This study shows the possibility of applying the fly ashes accumulated to date after the natural catastrophe that occurred on the island of La Palma, which would lead to the elimination of their accumulation and generate low-emission cement with good mechanical properties.

## 2. Materials and Methods

In this study, an analysis of the properties of volcanic material as an active mineral addition for the manufacture of cement was carried out. For this purpose, an extraction of volcanic material from two points of the island of La Palma, a collection of material close to the eruption of the volcano (fine ash) and two collections of material near the coastline (coarse ash and volcanic lava), were used. The material was processed by mechanical means through a crusher and impact mill, obtaining as a result two powdery materials with different degrees of fineness for each of the ashes studied and a powdery material from the volcanic lava.

An advanced characterisation study was done for each of the materials obtained, focusing on the fineness of the material, chemical composition, crystallography, and pozzolanic activity. Once the material was characterised, the evaluation of the volcanic material as an active mineral addition was carried out. For this purpose, 25% cement substitutions were made, and the fresh properties of the pastes and the pozzolanic capacity of the ashes were evaluated by means of different tests. Figure 1 shows a graph of the experimental methodology developed.

### 2.1. Raw Materials

In this section, the physical, chemical, and mineralogical properties of raw and processed volcanic ash and volcanic lava were studied to evaluate their pozzolanic potential and their effect as an additive in the development of new cements. 

In addition, two artificial pozzolanic materials, silica fume and fly ash, which are widely used in the cement industry, were studied as a reference, along with limestone filler.

#### 2.1.1. Ordinary Portland Cement

For the performance of this research, an ordinary commercial Portland cement of type CEM I 42.5R was used. A study of the main cement composition elements was carried out by fluorescence study. The main composition and density of OPC used is shown in Table 1.

#### 2.1.2. Fly Ash

The fly ash (FA) used in this study is a commercial artificial pozzolan used in the manufacture of cements. FA comes from the combustion of coal in power generation plants and is collected in filters by electrostatic precipitation. As can be seen in the laser granulometry, as shown in Figure 2, FA is the finest material analysed, with an average retained size of approximately 20 microns.

Table 2 shows that the real density of the FA has a value of 2.34 g/cm^3^, in addition to presenting a composition with high amounts of silicon, aluminium, and iron; these values are typical of coal fly ash [32]. Moreover, the reactive SiO_2_ value is higher than the 25% imposed by the standard.

Figure 3 shows the XRD pattern of the FA. As can be observed, the analysed FA shows a image in the diffractogram that indicates an important presence of amorphous phase as well as peaks of SiO_2_ crystalline found in different phases and mullite; this composition is coherent with that presented in other studies [33]. 

#### 2.1.3. Silica Fume

Silica fume (SF), or microsilica, is an inorganic product consisting of fine spherical particles formed from the reduction of quartz with carbon in the silicon metal and ferro-silicon manufacturing processes in electric arc furnaces. The dust produced is a byproduct collected in baghouses and silica dust collectors.

The silica fume analysed in this study has a particle average size of approximately 40 microns, as shown in Figure 2, as well as an actual density of 2.24 g/cm^3^. It is composed entirely of amorphous SiO_2_, as can be observed in Table 2 and in the XRD pattern in Figure 4. For this reason, silica fume has a great pozzolanic potential, which is observed with a SiO_2_ percentage higher than 60% and is widely applied in the manufacture of cements and concretes [34,35].

#### 2.1.4. Limestone Filler

A study of limestone filler, as a mineral addition without activity, was carried out to compare the effect of using volcanic material. It is a material of inorganic nature and mineral origin composed mainly of calcium carbonate (at least 75%), with a clay content of less than 1.2%. As shown in Table 2, its main composition is CaO.

#### 2.1.5. Volcanic Lava and Volcanic Ash

In this section, the physical, chemical, and mineralogical properties of the volcanic materials analysed are shown, and, in the following section, their pozzolanic potential compared to FA and SF is evaluated in order to determine the possibility of applying them in cementitious materials. 

Three volcanic materials were analysed: one sample of volcanic lava and two samples of pyroclasts. 

- Volcanic lava, from the solidified magma ejected by the volcano and collected from solidified lava flows close to the coastline, is called VL.

- Lapilli pyroclastic are particles between 2 and 64 mm in size ejected from the crater during ejection. Due to their size, lapilli pyroclastic precipitate by gravity in the areas near the crater, where the samples were collected, and are referred to as CVA (coarse volcanic ash).

- Ash type pyroclastic are particles smaller than 2 mm expelled during ejection; due to their size they can be deposited over long distances. They were collected near the coastline of the island and are referred to as FCV (fine volcanic ash).

Analysing the data shown in Table 2 for the three volcanic materials, it is observed that the densities of the volcanic ash vary between 2.30 g/cm^3^ and 2.90 g/cm^3^, with the lowest density in the FVA and the lowest in the CVA, due to the more compact granulometry of the fine ash, which gives them a higher density. Volcanic lava has an intermediate density value of 2.72 g/cm^3^; similar values have been shown in studies of volcanic ash from other eruptions. [15]

The three materials present a practically identical composition, due to the fact that they come from the same volcanic material in the interior of the earth, varying in the process of expulsion and subsequent deposit and cooling of the materials. VL, FVA, and CVA present a composition with silica as the major element, with values between 14%–19%, followed by Fe, with values between 8%–9%, Al and Ca, with values between 6%–8%, and Na and Mg, with values close to 3%.

However, the reactive SiO_2_ content is similar in both types of ash, in the order of 45%, but higher than 60% for volcanic lava, indicating a higher pozzolanic potential in this material.

FVA, CVA, and LV were studied by using X-ray diffraction (XRD). XRD data were collected at room temperature using Cu-Kα radiation (λ = 1.5406 Å) operated in the reflection geometry (θ/2θ). Data were recorded from 10° to 60° (2θ) with a step-size of 0.02. The X-ray tube was operated at 40 kV and 40 mA. Analysing the main component determined by X-ray fluorescense for the three volcanic materials, the XRD pattern shown in Figure 5, and the legend of the majority phases found (Table 3), it is observed that they were mainly composed of pyroxenes belonging to the inosilicate family, such as diopside and augetite, followed by feldspars of the tectosilicate family, where the presence of andesine, albite, and anorthoclase stand out. In addition, other crystalline phases were observed in the form of titanium oxides (rutile) and silicon oxide (quartz). Although the composition of volcanic materials depends on several factors, such as location and type of eruption, similar compositions have been found in volcanic ashes analysed by other authors [36,37].

The morphology of the volcanic material was determined with scanning electron microscopy (SEM), complemented with EDX to complete the compositional studies. A Hitachi S4800 electron microscope (Tokyo, Japan) was used for the morphology study. For the determination by energy dispersive spectroscopy (EDX) of the chemical composition of the samples, a Bruker Nano XFlash 5030 silicon drift detector was used.

Figure 6 shows the micrographs of FVA (a), CVA (b) and VL (c). A non-uniform microstructure is observed with the presence of larger angular particles in CVA and smaller ones in FVA. The presence of crystals was observed in all three volcanic materials analysed.

The existence of large quartz crystals, as observed in Figure 6, corresponds to the mineralogy of the volcanic material (Figure 5). The higher proportion of calcium and aluminum observed by XRD patterns (Table 4) would explain the formation of inosilicates and tectosilicates (Table 3) and corresponds with what has been observed by other authors who carried out analyses of volcanic material.

Volcanic lava is extracted from the lava flows by mechanical means, which involves obtaining particle sizes of several centimetres in diameter. Furthermore, volcanic ash (FVA and CVA) present coarser granulometry than FA and SF, as shown in Figure 7, which prevents their direct application as a mineral addition in the manufacture of new cements.

For this reason, a size reduction process is carried out on the samples to obtain the necessary particle size for the application as a mineral addition. The processing applied is as follows:(1)Drying of the material in an oven at 60 degrees Celsius.(2)Previous size reduction in a jaw crusher. Reduction in the initial fraction to a size of less than 4 mm (VL and CVA).(3)Grinding by impact mill with different abrasive loads and processing times.

The processes applied on volcanic ash and volcanic lava were two, from more abrasive (P1) to less abrasive (P2). This micronisation process aims to have a sufficient specific surface area to act as a cementitious material. The processes were carried out by introducing a determined quantity of material and abrasive load in a standardised friability test machine, subjecting them to a determined number of turns for their correct pulverisation.

After the size reduction process, five processed materials were obtained. The nomenclature of these materials is shown in Table 5.

#### 2.1.6. Evaluation of the Pozzolanic Potential of Raw Materials

Once the materials involved in the research have been analysed, a preliminary study is carried out to evaluate the pozzolanicity of the raw materials using the fixed lime method.

To study the pozzolanic activity of these materials, an accelerated method was used to measure the evolution of the material–lime reaction as a function of time. The test consisted of placing the different pozzolanic materials in contact with the saturated lime solution at 40 ± 1 °C for 3, 7, 28, and 90 days. At the end of this period, the CaO concentration in the solution was measured. The fixed lime (mM/L) was obtained from the difference between the concentration in the saturated lime solution and the CaO in the solution in contact with the sample at the end of the given period. The fixed lime value is a good indicator of the pozzolanic activity of the materials. It is higher as the amount of fixed lime increases. This method has been extensively described and applied by De Rojas and Frias, Rojas et al., and Frías et al. [38,39,40], allowing a preliminary evaluation of the pozzolan activity of raw materials with high reliability.

Figure 8 shows the lime absorption results for the two artificial pozzolanic materials and the three natural, volcanic pozzolanic materials. According to the results obtained, SF shows a high pozzolanic reactivity from the beginning of the test at 3 days, which is maintained up to 90 days. This is due to the high fineness presented by the silica fume samples together with their morphology mainly composed of amorphous silica.

The AF summarises a lime absorption that increases with time, reaching its maximum level at 90 days and presenting a 50% absorption with respect to the SF at 28 days. VL presents a similar behaviour to the AF, exceeding its pozzolanic activity by 60% at 28 days; however, it presents a similar activity at 90 days.

Finally, analysing the pozzolanic reactivity for volcanic ash, it is observed that the difference in particle size does not have a significant effect, presenting similar values. Compared to the rest of the values, it shows the lowest amounts of fixed lime, increasing to levels comparable with FA and VL, indicating that the pozzolanic activity of the ash is a long-term process from the beginning of the reaction.

### 2.2. Mix Proportions

In this section, the proportions of each material to be used to mix the mortars to be analysed were shown. Table 6 shows the dosages of each material, and the nomenclature of the mortar performed. Standardized sand (SNS) was used for the manufacture of the mortars, in accordance with UNE EN 196-1. However, because the amounts of materials included in the mixtures are too numerous to be tabulated, and the percentage of each material added is the same, replacing 25% of cement by each pozzolan, the materials derived from the volcanic ashes (FVA-NP, FVA-P1, FVA-P2, CVA-NP, CVA-P1, CVA-P2) are referred to as FVA, CVA, and LF. 

### 2.3. Test Procedures

The tests carried out to evaluate the pozzolanicity of the volcanic material are shown below.

#### 2.3.1. Pozzolanicity and Frattini Tests (UNE_EN 196-5:2011)

Pozzolanicity is a test carried out on cement substitutes. By performing this test, it is possible to quantify the amount of calcium oxide that a material is capable of fixing. To determine the pozzolanicity, the material to be tested is immersed in a saturated calcium oxide solution, and the levels of calcium oxide absorbed by the sample were measured. The results were shown as the percentage of calcium oxide fixed in the sample out of the total calcium oxide in the solution.

The Frattini test, similar to the pozzolanicity test, is carried out on cements and mixtures of cements with substitutes. In accordance with the standard, the cement is immersed in a solution in which, after 8 and 15 days, the amount of hydroxyl ions and the amount of calcium oxide that has been absorbed by the sample were evaluated. To evaluate its pozzolanic capacity, the values obtained were presented in a graph that plots the concentrations of hydroxyl ions against the concentrations of calcium oxide on its axes. 

The standard presents a curve that divides the graph into two zones. If the point resulting from the test is below this curve, the material is potentially pozzolanic. If it is above, the material is not pozzolanic.

This is a method for evaluating the pozzolanicity of pozzolanic cement, which, therefore, serves to evaluate the pozzolanic behaviour of a material when mixed with cement in different proportions. To evaluate the effect of volcanic material (FVA, CVA, and VL), cement/volcanic material mixtures were prepared. 

To test the effect of volcanic material in a cement, cement/volcanic material mixtures were prepared in 75/25 proportions. The Portland cement used as a reference was CEMI/42.5R, which has a clinker content equal to or greater than 95%, so it can incorporate additional components up to 5%. 

#### 2.3.2. Resistant Activity Index (UNE_EN 196-1:2018)

The determination of the pozzolanic activity index in Portland cement is defined as the ratio between the maximum load supported by the test mortars (standard cement with added pozzolan) and the maximum load supported by the standard mortars (standard cement), expressed in percentage terms. In other words, it is the variable that allows a pozzolan to be classified for use in the cement production process, a value that is internationally considered to be at least 75%. The existing physical–mechanical methods for the determination of this index require waiting 28 days from the completion of the test expressed in the standard, to stipulate whether a material has acceptable pozzolanic properties for use.

Additionally, to evaluate the pozzolanic activity of the material, a study of the resistant activity index was carried out in accordance with the UNE 450-1 standard. It should be noted that this standard refers to the use of fly ash; its application is not mandatory for natural pozzolans, and, therefore, the established minimum compressive strength requirements do not have to be met.

This method includes the determination of compressive and flexural strengths, according to UNE 196-1 of prismatic specimens, of dimensions 40 mm × 40 mm × 160 mm, prepared with 75% of the test cement plus 25% by mass of volcanic ashes. The specimens were kept in the mould in a humid atmosphere for 24 h, and, after demoulding, the specimens were immersed in water until the strength tests were performed at the required age, in this case at 7, 28, and 90 days.

#### 2.3.3. Setting Time and Volumetric Expansion (UNE-EN 196-3:2017)

Setting time is expressed in two values, the initial setting time and the final setting time. The initial setting time refers to the number of minutes that elapse from the time the cement comes into contact with water until the cement paste begins to lose its plasticity. The final setting time is expressed as the number of minutes that have elapsed, since the water comes into contact with the cement, until the cement paste loses its plasticity totally and is completely hardened.

To quantify these times in a standardised way, the test is performed according to EN 196-3 standard. The Vicat apparatus is used in this test, in which the degree of penetration of its needles will determine the hardening of the cement paste, thus being able to determine the initial and final setting times.

Volumetric expansion determines the change in volume that the cement undergoes during hardening. These values are relevant because they determine the soundness of cement. To determine the volumetric expansion, the cement paste is tested according to EN 196-3, and, through the measurements on the Le Chatelier needles, we can determine the volumetric expansion of the cement under test.

## 3. Results and Discussion

### 3.1. Pozzolanity and Frattini Tests

Figure 9 shows the results obtained for the [CaO] and [OH^−^] concentrations of each of the mixtures analysed at 8 and 15 days according to the standardised test. The results were compared with the portlandite solubility curve. 

Figure 9 shows the values of [Ca]^2+^ and [OH^−^] oxides, which decrease in solution as a consequence of the depletion of calcium hydroxide, after the pozzolanic reaction of each of the mixes analysed at 8 and 15 days, according to the standardised test. The results were compared with the portlandite solubility curve. 

The analysed mixture is considered to comply with the test, i.e., to be pozzolanic, when the concentration of calcium ions is lower than the saturation concentration indicated by the reference curve. It was observed that all ash mixtures analysed were above the curve, unlike the results obtained in other studies [41], in which volcanic ashes showed high pozzolanicity as well as fly ash and silica fume [42,43].

The crushing processing of the volcanic ash led to an improvement in the pozzolanic capacity of the material; in the short term, the material was not considered to be pozzolanic, but the values were close to the solubility curve. Previous studies showed that mechanical activation of ashes increases the reactivity of the pozzolanic material [44].

As shown by other authors, crushing volcanic material to be used as a supplementary cementitious material improved the pozzolanic properties of the mixes [45,46,47]. The approximation to the solubility curve of the crushed material is due to the fact that higher amounts of calcium silicate hydrate (C-S-H) and calcium aluminate silicate hydrates (C-A-S-H) gel phases were formed, and finer sizes of the material lead to a higher amount of these phases.

### 3.2. Resistant Activity Index

It can be seen in Table 7 that the processed volcanic ash improves its resistance compared to unprocessed volcanic ash; that is, the degree of fineness has a very relevant influence on the resistance obtained. If FVA-2 and CVA-2 were compared with the mixture in which FA was used, it remains slightly below, not reaching 85% in the case of volcanic ash. However, a much higher increase than the mixture with LF is obtained, which allows one to think that they are usable in the manufacture of cement and as a mineral addition to concrete.

In Figure 10, only the five most significant mixtures have been included. It is clearly observed how there is a more relevant increase in resistance in the two samples made with processed volcanic ash (CVA-2 and FVA-2), compared to the control or in the mixture made with FA. This fact indicates that the volcanic ash gradually increases its resistance over time, with its growth being greater after 28 days, compared to the case of mixtures made with a conventional cement. This is due to an increase in pozzolanicity at 90 days, as observed in Figure 8, which contributes higher strength to the mortars made with FVA and CVA. On the other hand, in the mixture made with LF, it does not show significant growth after day 28 because it is a mineral addition with little pozzolanic activity. 

Therefore, based on these results, it can be concluded that volcanic ashes processed with an adequate degree of fineness can be used in the manufacture of cement, presenting a higher reactive silica content at 42%, which is very important for the validation of these volcanic materials as a cement substitute. Lastly, it should be noted that the unprocessed volcanic ash presented similar results to the mortars made with LF at 90 days, with the behaviour being less than 28 days and the growth of resistance being between 28 and 90 days. For example, in both mixtures, FVA-NP and CVA-NP, it is possible to observe the increase in resistance between 28 days and 90 days, going from 31.1 MPa to 37.8 MPa in CVA-NP (increase of 21.5%), and from 31.6 MPa to 37.1 MPa in FVA-NP (increase of 17.4%). 

These results clearly show that volcanic ash can be used as SCM, and, although it can be processed, improving behaviour, it can be applied in the manufacture of cements and as a mineral addition in concrete manufacturing.

### 3.3. Setting Time and Volumetric Expansion

Table 8 shows the results obtained for the initial and final setting times and volumetric expansion for all mortars. As can be observed, the OPC initial and final setting times were 105 and 190 min, respectively, consistent with high percentages of clinker and rapid hardening cement. 

The addition of LF retarded both setting times. On the contrary, the addition of FA implies a lengthening of both times. This behaviour has been extensively studied and described for decades [48,49,50].

Analysing the results for the VL, FVA, and CVA samples, it is observed that the addition of these materials and their different processing slightly decrease the initial setting time as well as more noticeably reduce the final setting time; however, the times between the different volcanic materials remain practically stable. Other studies describe an opposite behaviour after adding these materials, with slight increases in setting times; however, due to the different origins, compositions, and properties of the volcanic materials, different effects can be observed [15]. 

Concerning the results for the determination of volumetric expansion, all materials show values below the limits for cementitious specifications according to EN 197-1. The addition of SF has no significant effect on the expansion of the cementitious pastes [49], although the addition of FA does lead to an increase in expansion. The addition of volcanic material slightly reduces the volumetric expansion of the cementitious pastes and, as with the setting times, the values for all volcanic materials are similar. This decrease in expansion with respect to the mortars manufactured with FA may be mainly due to the increased absorption of the mortar pastes manufactured with FA [51,52]. The FVA and CVA samples presented low absorption (Table 2); therefore, the manufactured mortars presented low dimensional changes at early ages, due to the fact that there are no large pores or occluded water in the mortars that could modify their dimensions at initial curing ages.

## 4. Conclusions

In the present study, the effect of applying volcanic material (pyroclasts and volcanic lava) from the eruption of the Cumbre Vieja volcano in La Palma, Spain, as an active mineral addition for the manufacture of pozzolanic cements, is analysed. In addition, silica fume and fly ash from coal combustion were analysed as pozzolanic material references. After studying the physical, chemical, and mineralogical properties of volcanic materials and their application in mortars, the following conclusions are drawn:

- Volcanic material (fine ash, coarse ash, and lava) is mainly composed of SiO_2_, Al_2_O_3_, Fe_2_O_3_, and CaO. A natural pozzolan is essentially composed of reactive silicon dioxide (SiO_2_), aluminium oxide (Al_2_O_3_), and iron oxide (Fe_2_O_3_). Therefore, the material has suitable characteristics to be used as natural pozzolanic material as SCM.

- The three materials analysed (coarse ash, fine ash, and lava) have reactive silicon dioxide values well above the 25% required by the UNE-EN 197-1 standard for the application of natural pozzolan in the manufacture of cement. This demonstrates that their use is viable and complies with the minimum requirements established.

- The pozzolanicity study showed that the volcanic lava presented high pozzolanicity at early ages; however, the volcanic ash evolved more positively, obtaining high pozzolanicity at 90 days. This is a positive fact, since natural pozzolans cannot be evaluated only in the short term: it is necessary to evaluate their mechanical behaviour in the medium and long term.

- The unprocessed volcanic ash showed a resistance in the 28-day resistant activity index test that was lower than the rest of the SCM studied in this work, but the increase in resistance between 28 and 90 days was much higher, obtaining up to a 21.5% increase in resistance in the sample in the mortar mix made with CVA-NP.

- A relevant increase was observed in resistance in the processed volcanic ash, and the mixtures made with them increase their resistance over time, so the increase between 28 and 90 days was very relevant. 

- In the long term (90 days), the compressive strength results of mortars manufactured with FVA and CVA increased considerably, exceeding the results obtained in the LF mixtures.

- In the long term, it is demonstrated that unprocessed and crushed volcanic ash can be used as a natural pozzolan for the manufacture of cement, obtaining higher results than a mortar made with limestone filler.

In view of the results, the pozzolanic potential of the volcanic ash from the La Palma eruption is feasible for the manufacture of cement, and it is possible to apply substitution percentages of SCM of up to 25%. This application shows the environmental and social benefits in relation to the volcanic process that occurred in 2021 on the island of La Palma, Spain, due to the large volume of fly ash generated during the eruption of the volcano.

## Figures and Tables

**Figure 1 materials-15-06305-f001:**
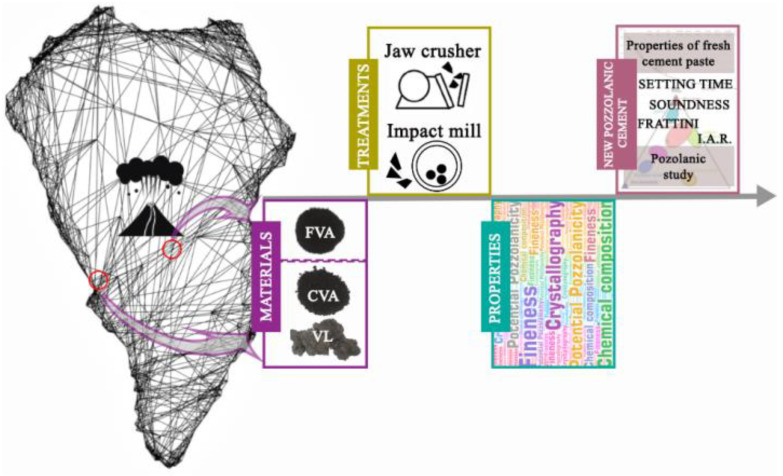
Experimental scheme.

**Figure 2 materials-15-06305-f002:**
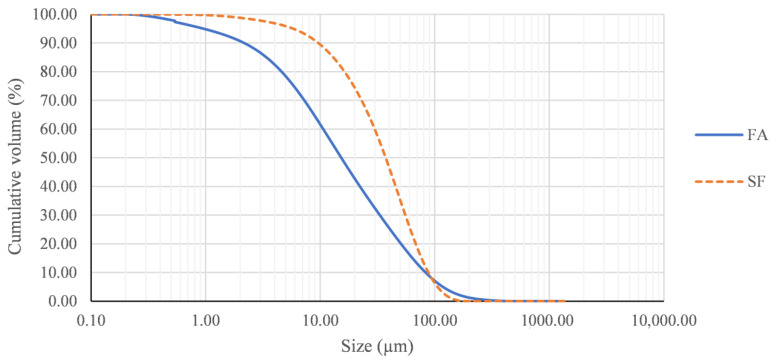
Particle size distribution of fly ash (FA) and silica fume (SF).

**Figure 3 materials-15-06305-f003:**
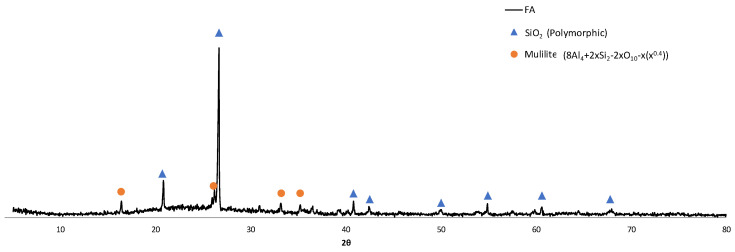
XRD pattern of FA.

**Figure 4 materials-15-06305-f004:**
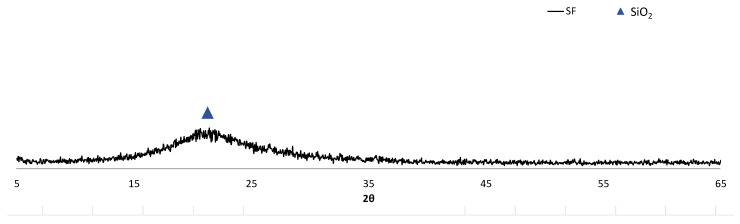
XRD pattern of SF.

**Figure 5 materials-15-06305-f005:**
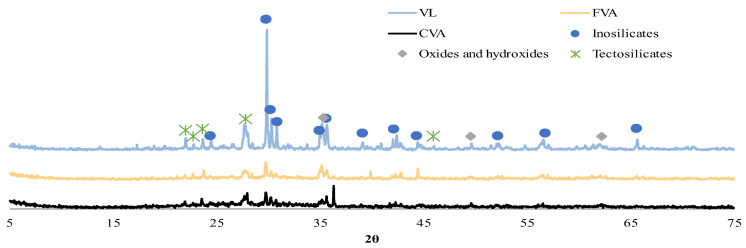
XRD patterns of VL, FVA, and CVA.

**Figure 6 materials-15-06305-f006:**
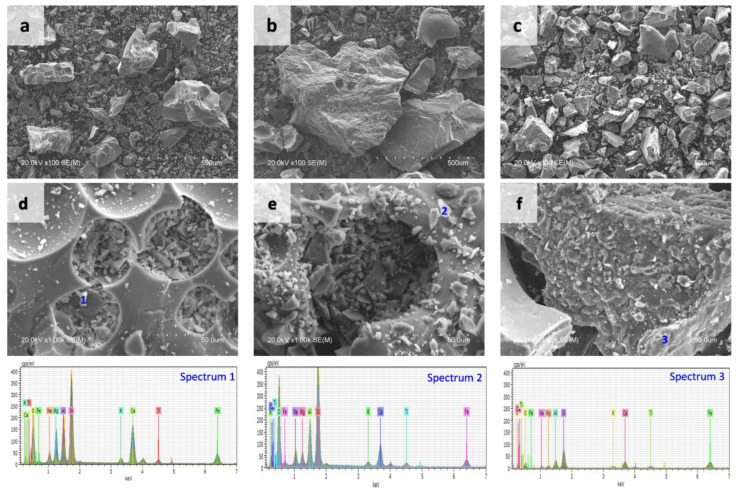
SEM micrographs of volcanic materials: (**a**) FVA 500 µm; (**b**) CVA 500 µm; (**c**) VL 500 µm; (**d**) FVA 50 µm; (**e**) CVA 50 µm; and (**f**) VL 50 µm.

**Figure 7 materials-15-06305-f007:**
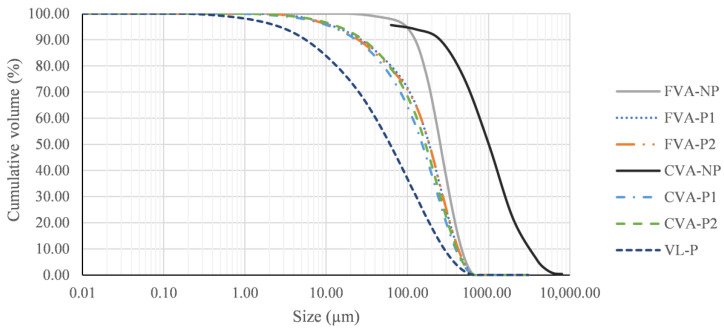
Particle size distribution of raw and processed VA and VL.

**Figure 8 materials-15-06305-f008:**
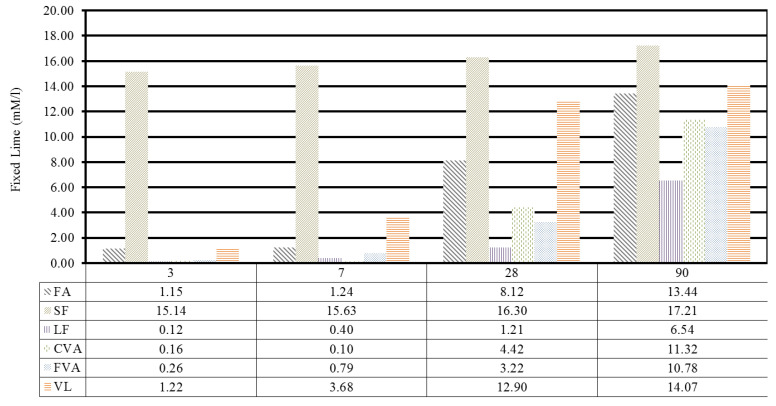
Fixed lime in pozzolans over time.

**Figure 9 materials-15-06305-f009:**
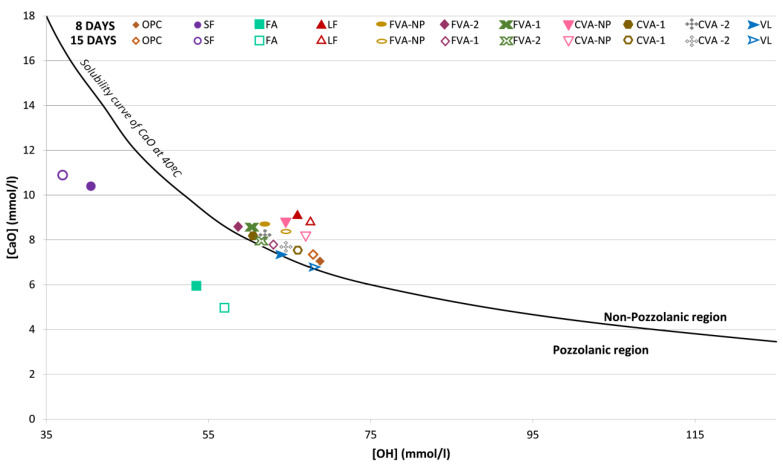
Results of Frattini test at 8 and 14 days.

**Figure 10 materials-15-06305-f010:**
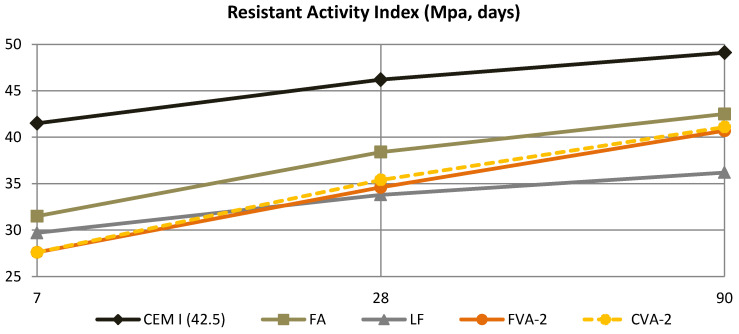
Progress in resistance activity index.

**Table 1 materials-15-06305-t001:** OPC main components and density.

Main Components FRX (%)	CaO	SiO_2_	SO_3_	Al_2_O_3_	Fe_2_O_3_	MgO	K_2_O	Na_2_O	TiO_2_	Density (g/cm^3^) (UNE EN 196-6)
CEM I 42.5R	66.22	17.95	5.44	4.25	2.89	1.36	1.12	0.39	0.19	3.11

**Table 2 materials-15-06305-t002:** Physicochemical properties of the raw pozzolans.

	Fly Ash (FA)	Silica Fume (SF)	Limestone Filler (LF)	Fine Volcanic Ash (FVA)	Coarse Volcanic Ash (CVA)	Volcanic Lava (VL)	Standard
Real density (g/cm^3^)	2.34	2.24	2.67	2.9	2.3	2.72	EN 1097-6
Water absortion (%)	-	-	-	0.38	0.44	1.91	
Reactive SiO_2_ (%)	12.2	64.9	1.52	44.3	42.8	61.9	EN 80225
Organic matter content (%)	0.00	0.00	0.00	0.00	0.00	0.003	UNE 103204
Water-soluble sulphate (% SO_3_)	0.26	0.38	0.00	0.00	0.00	0.0001	EN 196-2
Main components EDX/EDS (%)							
Na	0.35	0.08	0.21	2.91	3.07	2.27	
P	0.22	0.07	0.01	0.38	0.33	0.26	
Si	15.8	35.4	0.66	19.14	19.04	14.73	
Ca	1.83	0.63	38.98	7.64	8.31	6.75	
Al	9.92	0.38	0.02	7.82	7.76	5.67	
S	0.16	0.16	0.04	0.08	0.09	0.05	
K	2.5	0.43	0.02	1.54	1.54.	1.34	
Mg	0.81	0.26	0.39	3.11	3.52	3.03	
Fe	3.69	0.18	0.01	9.12	9.75	8.08	

**Table 3 materials-15-06305-t003:** Mineralogical phases in volcanic materials.

Oxides and Hydroxides
Magnetite (Fe_3_O_4_)
Quartz (SiO_2_)
Rutile (TiO_2_)
**Inosilicates**
Diopside (Ca Fe_0.205_ Mg_0.895_ O_6_ Si_1.9_)
Augite (Ca Fe_0.25_ Mg_0.74_ O_6_ Si_2_)
**Tectosilicates**
Andesine (Al_0.735_ Ca_0.24_ Na_0.26_ O_4_ Si_1.265_)
Bytownite (Al_7.76_ Ca_3.44_ Na_0.56_ O_32_ Si_8.24_)
Labradorite (Al_0.81_ Ca_0.325_ Na_0.16_ O_4_ Si_1.19_)
Sanidine (Al_1.04_ Ca_0.04_ K_0.65_ Na_0.31_ O_8_ Si_2.96_)
Albite (Al Na O_8_ Si_3_)
Anorthoclase (Al_1.1_ Ca_0.1_ K_0.27_ Na_0.63_ O_8_ Si_2.9_)

**Table 4 materials-15-06305-t004:** Chemical composition of volcanic materials performed by energy dispersive spectroscopy determines the (wt%).

	SiO_2_	Al_2_O_3_	Fe_2_O_3_	MgO	CaO	Na_2_O	SO_3_	K_2_O	TiO_2_
FVA	40.72	18.34	12.82	4.69	9.81	7.11	0.01	2.04	4.06
CVA	40.65	18.11	13.46	1.77	17.85	2.26	0.01	1.91	3.89
VL	36.58	16.74	25.79	0.36	13.54	1.05	0.01	1.09	4.82

**Table 5 materials-15-06305-t005:** Nomenclature volcanic material.

Description	Nomenclature
Non-Processed Pulverized Fine Volcanic Ash	FVA-NP
Pulverized Fine Volcanic Ash Implementing Process 1	FVA-1
Pulverized Fine Volcanic Ash Implementing Process 2	FVA-2
Non-Processed Pulverized Coarse Volcanic Ash	CVA-NP
Pulverized Coarse Volcanic Ash Implementing Process 1	CVA-1
Pulverized Coarse Volcanic Ash Implementing Process 2	CVA-2
Pulverized Volcanic Lava	VL-P

**Table 6 materials-15-06305-t006:** Dosages of mortar made in laboratory.

Mixture	Description	Dosages (g)								
		SNS	OPC	SF	FA	LF	FVA	CVA	VL	Water
OPC	OPC—Cem I	1350	450		-	-	-	-	-	225
SF	10% Silica Fume	1350	450	112.5	-	-	-	-	-	225
FA	25% Fly Ash Addition	1350	337.5		112.5	-	-	-	-	225
LF	25% Limestone Filler	1350	337.5		-	112.5	-	-	-	225
FVA-NP	25% Fine Volcanic Ash Addition; Non-Processed	1350	337.5		-	-	112.5	-	-	225
FVA-1	25% Fine Volcanic Ash Addition; Process 1	1350	337.5		-	-	112.5	-	-	225
FVA-2	25% Fine Volcanic Ash Addition; Process 2	1350	337.5		-	-	112.5	-	-	225
CVA-NP	25% Coarse Volcanic Ash Addition; Non-Processed	1350	337.5		-	-	-	112.5	-	225
CVA-1	25% Coarse Volcanic Ash Addition; Process 1	1350	337.5		-	-	-	112.5	-	225
CVA-2	25% Coarse Volcanic Ash Addition; Process 2	1350	337.5		-	-	-	112.5	-	225
VL	Volcanic Lavage Addition	1350	337.5		-	-	-	-	112.5	225

**Table 7 materials-15-06305-t007:** Results of compressive strength in resistant activity index.

	Compressive Strength MPa (Age)			% Regarding Control 28D	% Regarding Control 90D	Resistance Increase 28–90 days
	7	28	90			
CEM I (42.5)	41.5	46.2	49.1	-	-	6.4%
SF	35.6	42.4	45.5	91.8%	92.6%	7.3%
FA	31.5	38.4	42.5	83.1%	86.5%	10.7%
LF	29.7	33.8	36.2	73.2%	75.7%	7.1%
FVA-NP	26.7	31.6	37.1	68.4%	75.5%	17.4%
FVA-1	26.2	31.5	36.2	68.2%	73.7%	14.9%
FVA-2	27.6	34.6	40.7	74.9%	82.8%	17.6%
CVA-NP	19.7	31.1	37.8	67.3%	76.9%	21.5%
CVA-1	24.3	33.7	38.6	72.9%	78.5%	14.5%
CVA-2	27.6	35.4	41.1	76.6%	83.6%	16.1%
VL	25.5	33.8	37.1	73.2%	75.5%	9.8%

**Table 8 materials-15-06305-t008:** Setting time and volumetric expansion of mortar.

Mixture	Setting Time (min)		Expansion (mm)
	Initial	Final	
OPC	105	190	1.50 mm
SF	110	180	1.40 mm
FA	125	235	2.10 mm
LF	85	175	1.30 mm
FVA-NP	90	140	1.0 mm
FVA-P1	105	155	0.9 mm
FVA-P2	90	140	1.1 mm
CVA-NP	90	140	1.0 mm
CVA-P1	95	140	0.9 mm
CVA-P2	90	130	1.0 mm
VL-P	100	135	0.8 mm

## Data Availability

Not applicable.

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
