# Peer review of "Effect of Processed Volcanic Ash as Active Mineral Addition for Cement Manufacture"

_materials, 2022, doi:10.3390/ma15186305_

Round 1

Reviewer 1 Report

The comments are in the file.

Author Response

Dear Ms. Jayda Zhang:

Attached please find our revised manuscript ID materials-1868848 entitled:

“EFFECT OF PROCESSED VOLCANIC ASH AS ACTIVE MINERAL ADDITION FOR CEMENT MANUFACTURE”.

We are grateful for the constructive comments by yourself, which we believe have allowed us to substantially improve our manuscript.

We have thoroughly revised the manuscript. I would like to first address the principle concerns (in blue) and indicate how we have dealt with them:

Reviewers' comments:

Reviewer #1:

  1. Some quantitative experimental results are required in the abstract.

Thank you for your comments. This aspect has been clarified (lines 21-23)

  1. Although a good literature review is provided, only two paper published in the current year (2022) is mentioned in the manuscript. In this regard, some updates are necessary since tens of papers are published in this discipline every year.

Thank you for your comment. Updated papers from the year 2022 have been entered into the manuscript. The studies referred to are listed below.

Khan, K., Amin, M. N., Usman, M., Imran, M., Al-Faiad, M. A., & Shalabi, F. I. (2022). Effect of Fineness and Heat Treatment on the Pozzolanic Activity of Natural Volcanic Ash for Its Utilization as Supplementary Cementitious Materials. Crystals, 12(2), 302.

Shahjalal, M., Rahman, J., Haque, A. F., Habib, L., Jalal, K. B., & Rahman, M. M. (2022). Effect of Partial Replacement of Cement with Volcanic Ash on Mechanical Behaviour of Mortar. In Proceedings of the 5th International Conference on Sustainable Civil Engineering Structures and Construction Materials (pp. 33-44). Springer, Singapore.

Churata, R., Almirón, J., Vargas, M., Tupayachy-Quispe, D., Torres-Almirón, J., Ortiz-Valdivia, Y., & Velasco, F. (2022). Study of Geopolymer Composites Based on Volcanic Ash, Fly Ash, Pozzolan, Metakaolin and Mining Tailing. Buildings, 12(8), 1118.

ABDULMALEK, N., & Chakkamalayath, J. (2022). Cost-Benefit Analysis of Vibrated Cement Concrete and Self-Compacting Concrete Containing Recycled Aggregates and Natural Pozzolana. Journal of Engineering Research.

Cultrone, G. (2022). The use of Mount Etna volcanic ash in the production of bricks with good physical-mechanical performance: Converting a problematic waste product into a resource for the construction industry. Ceramics International, 48(4), 5724-5736.

Alqarni, A. S. (2022). A comprehensive review on properties of sustainable concrete using volcanic pumice powder ash as a supplementary cementitious material. Construction and Building Materials, 323, 126533.

  1. The introduction section is very limited. A rich but brief literature review must be added about the effects of each parameter that was used in the study.

Thank you for your comment. The literature review has been expanded. The authors consider that there is a broad view concerning the effects of volcanic ash application on different aspects studied.

  1. It is good to cite this very recent and related paper in the introduction, and literature and to support your discussion: "The Effect of Openings on the Performance of Self-Compacting Concrete with Volcanic Pumice Powder and Different Steel Fibers", 2022

Thank you for your contribution, we have used the proposed paper (reference number 30) to expand on the discussion in the Introduction (lines 71-74).

  1. The novelty of the article is not clear when rich literature is available on this topic. Based on the required up-to-date literature survey, the authors must clearly highlight the novel points and significance of the introduced work.

Thank you for your comment. A description of the novelty of the work has been included. Volcanic ashes have different properties depending on their origin, so a study is always considered necessary for their possible application. In lines 85-90 the novelty of this study has been explained.

  1. Figure 1 is not clear, let alone when printed in black and white. It is recommended that the layout of the graphs is changed to something clearer.

Figure 1 in black and white is shown below. The authors believe it is clearly visible.

  1. Were these values calculated in this study? If it is "yes," please mention that in the text, and if it is "no," it is preferable to mention the source of these values.

Thank you for your comment; it has been clarified in the text that the analysis of the OPC composition was carried out in the laboratory.

  1. Figure 9 is not clear in color, let alone when printed in black and white. It is recommended that the layout of the graphs is changed to something clearer, e.g., by using different colors or using markers.

Thank you for your comment, the figure has been replaced.

All abbreviations used in the text have been explained, as well as corrected comments in relation to the UNE standards used.

After complete the revision process, we hope that the revised manuscript does now fully meet the criteria and conditions for publication in Journal of Materials. Thank you very much for your efforts concerning our manuscript. 

Yours sincerely,

Ph D. Francisco Agrela

University of Córdoba

Reviewer 2 Report

    REVIEWER’S COMMENTS

The manuscript presents the study on the cementing and pozzolanic reaction characteristics of both unprocessed and processed (crushing) volcanic materials. The outcome of the study shows that volcanic materials processed by crushing are potential materials for cement manufacturing to improve the pozzolanic behaviour of cement. However, the authors need to address the following issues.

1. In ‘The introduction section’, merge paragraphs 1, 2, and 3 to form one paragraph.

2. In Line 42, separate and elaborate on the lumped references [8-12]  by providing some insights on the outcome of the studies individually.

3. Remove all the commas after the et al. in all your references. For example, Dingwell et al., [7] should be written as Dingwell et al. [7].

4. Remove double spacing in Line 55 before ‘Serra et al. [18]'

5. The sentence structure of the aim of the study is too complex and ambiguous as shown in Line 75-81 of the manuscript. Try breaking the long sentence into two sentences.

6. In the 'Materials and method section,' some vital information is missing. For example, the information on the particle size distribution of the volcanic materials ( fine ash, coarse ash, and volcanic lava) needs to be included.

7. In 'The materials and method section, merge paragraphs I and 2. Also, merge paragraphs 3 and 4.

8. Line 133, Fig. 3, correct the spelling of pattern. 'patter' should be written as 'patterns'.

9. To ensure each abbreviation is defined at first mention, move 2.1.5 forward to become 2.1.2 while you make 2.1.2 to become 2.1.5.

10. In 2.1.4, join paragraphs 1 and 2 together. You can also add more information to this sub-section if possible.

 11. The XRF result in Table 2 doesn't look correct. The correct XRF result should give percentages of oxides present in the materials instead of the elemental composition of the material as presented by the author. The author should correct this by indicating the result in table 2 as the EDX/EDS result of the material instead of the XRF result or by doing a proper XRF analysis.

12. Line 129, what is the meaning of the word 'halo'? 

 13. Line 131, remove the full stop after 'studies'

 14. Line 194, move the full stop after 'authors' to be after the reference '[29,39]'.

 15. The author should use Grammarly software to eliminate grammatical errors contained in the manuscript such as the use of 'is' for 'are', 'are' for 'were', etc.

 16. Lines 134-135, cite the references correctly by removing 'research group' and indicating the authors of the work cited.

 17. In Table 5, indicate what 'SNS' stands for. Also, remove the word 'Serie' from the Table 5 heading.

 18. Line 274, put the appropriate number for the citation in the place of '[REF].

19. Line 347, indicates what C-S-H and C-A-S-H stand for.

20. In Table 6, change all the commas in between number values to decimal. Effect this changes throughout the manuscript. For example '41,5' should be written as '41.5'

21. Lines 380-385, the authors should explain the mechanisms of increase in resistance for materials containing volcanic ash. State the reason(s) or what caused the increase in resistance. Also, indicate what led to the result obtained for other materials (FA and SF). Do the same for results in Lines 422-427.

 22. Line 397, add units o the value of the resistance. For example, ' 37.1' should be written as '37.1 MPa.

 23. The number in the chemical formula of the oxides in the 'Conclusion section) should be written as a subscript. For example, SiO2 should be written SiO2

 24. Lines 461- 465: the sentence is too complete and vague, try splitting the sentence into two sentences for better understanding.

Author Response

Dear Ms. Jayda Zhang:

Attached please find our revised manuscript ID materials-1868848 entitled:

“EFFECT OF PROCESSED VOLCANIC ASH AS ACTIVE MINERAL ADDITION FOR CEMENT MANUFACTURE”.

We are grateful for the constructive comments by yourself, which we believe have allowed us to substantially improve our manuscript.

We have thoroughly revised the manuscript. I would like to first address the principle concerns (in blue) and indicate how we have dealt with them:

Reviewers' comments:

Reviewer #2:

The manuscript presents the study on the cementing and pozzolanic reaction characteristics of both unprocessed and processed (crushing) volcanic materials. The outcome of the study shows that volcanic materials processed by crushing are potential materials for cement manufacturing to improve the pozzolanic behaviour of cement. However, the authors need to address the following issues.

  1. In ‘The introduction section’, merge paragraphs 1, 2, and 3 to form one paragraph.

Thank you for your comment. The paragraphs have been merged.

  1. In Line 42, separate and elaborate on the lumped references [8-12]  by providing some insights on the outcome of the studies individually.

Thank you for your comment, the introduction has been modified to take this into account.

  1. Remove all the commas after the et al. in all your references. For example, Dingwell et al., [7] should be written as Dingwell et al. [7].

Thank you. It has been corrected throughout the manuscript.

  1. Remove double spacing in Line 55 before ‘Serra et al. [18]'

Thank you, the double spacing has been removed

  1. The sentence structure of the aim of the study is too complex and ambiguous as shown in Line 75-81 of the manuscript. Try breaking the long sentence into two sentences.

Thank you for your comment. The aims of the work have been rewritten to make them clearer to readers.

  1.  In the 'Materials and method section,' some vital information is missing. For example, the information on the particle size distribution of the volcanic materials ( fine ash, coarse ash, and volcanic lava) needs to be included.

Thank you for your comment. Figure 6 shows the particle size of fine volcanic ash (FVA-NP) and coarse volcanic ash (CVA-NP). The particle size of lava has not been included because large portions of lava with a size greater than 100 mm were processed in the laboratory as can be seen in the attached figure, therefore, it is not considered appropriate to introduce its particle size.

  1. In 'The materials and method section, merge paragraphs I and 2. Also, merge paragraphs 3 and 4.

Thank you for your comment, the paragraphs have been merged.

  1. Line 133, Fig. 3, correct the spelling of pattern. 'patter' should be written as 'patterns'.

Thank you, it has been corrected

  1. To ensure each abbreviation is defined at first mention, move 2.1.5 forward to become 2.1.2 while you make 2.1.2 to become 2.1.5.

Thank you, it was not necessary to change the structure of the article. Following your indications and those of other reviewers, each of the abbreviations in the tables has been explained.

  1.  In 2.1.4, join paragraphs 1 and 2 together. You can also add more information to this sub-section if possible.

Thank you for your suggestion. The authors consider that the information provided is sufficient, since there is an extensive bibliography on the properties of limestone filler.

  1.  The XRF result in Table 2 doesn't look correct. The correct XRF result should give percentages of oxides present in the materials instead of the elemental composition of the material as presented by the author. The author should correct this by indicating the result in table 2 as the EDX/EDS result of the material instead of the XRF result or by doing a proper XRF analysis.

Thank you very much for your comment, the error has been corrected in the table.

  1. Line 129, what is the meaning of the word 'halo'? 

Thank you for your comment, the word has been replaced to better understand the sentence.

  1.  Line 131, remove the full stop after 'studies'

Thank you, it has been corrected.

  1.  Line 194, move the full stop after 'authors' to be after the reference '[29,39]'.

Thank you, it has been corrected.

  1.  The author should use Grammarly software to eliminate grammatical errors contained in the manuscript such as the use of 'is' for 'are', 'are' for 'were', etc.

Thank you, the text has been corrected throughout the manuscript.

  1.  Lines 134-135, cite the references correctly by removing 'research group' and indicating the authors of the work cited.

Thank you, it has been corrected

  1.  In Table 5, indicate what 'SNS' stands for. Also, remove the word 'Serie' from the Table 5 heading.

Thank you for your comment. The definition of SNS has been specified in the text and the term "series" has been removed from the table.

  1.  Line 274, put the appropriate number for the citation in the place of '[REF].

Thank you, it has been corrected.

  1. Line 347, indicates what C-S-H and C-A-S-H stand for.

Thank you, it has been defined in the manuscript

  1. In Table 6, change all the commas in between number values to decimal. Effect this changes throughout the manuscript. For example '41,5' should be written as '41.5'

Thank you, this error has been corrected throughout the manuscript.

  1. Lines 380-385, the authors should explain the mechanisms of increase in resistance for materials containing volcanic ash. State the reason(s) or what caused the increase in resistance. Also, indicate what led to the result obtained for other materials (FA and SF). Do the same for results in Lines 422-427.

Thank you for your comment. In the text there is an explanation of why each of the phenomena that occur in the properties of the manufactured mortars.

  1.  Line 397, add units o the value of the resistance. For example, ' 37.1' should be written as '37.1 MPa.

Thank you, it has been corrected in the manuscript.

  1. The number in the chemical formula of the oxides in the 'Conclusion section) should be written as a subscript. For example, SiO2 should be written SiO2

Thank you, it has been corrected throughout the manuscript.

  1.  Lines 461- 465: the sentence is too complete and vague, try splitting the sentence into two sentences for better understanding.

Thank you, the paragraph has been rewritten

After complete the revision process, we hope that the revised manuscript does now fully meet the criteria and conditions for publication in Journal of Materials. Thank you very much for your efforts concerning our manuscript. 

Yours sincerely,

Ph D. Francisco Agrela

University of Córdoba

Reviewer 3 Report

In the review of the manuscript titled: Effect of processed volcanic ash as active mineral addition for cement manufacture, overall description is good and the manuscript is written very well. I would like to see this article publish but after some minor modification as follow,

1.      Overall, manuscript must be revised throughout as superscripts and sub-scripts must be checked like;

i-                    Page 2, line 77, “CO2”.

ii-                  Page 4, line 122, “g/cm3”

iii-                Figure 3, within the figure formula of multilite

iv-                Page 5, line 140 and 141, “SiO2”

v-                  Page 6 line 170, 171

vi-                Figure 4, Inside the “Figure formula for SiO2”,

vii-              Page 15, second paragraph of conclusion. etc.,

2.      In the introduction portion please cite some references for literature review of your material upto date.

3.      In the section of method please add the specifications of the XRD machine with company and model.

4.       In my suggestion, it’s better to prepare the origin diagrams (Figure 3, 4 and 5) more attractive by adding the Y axis of intensity (a.u.) as well.

5.      The authors are requested to add the SEM images with their EDS of all three samples of 1- VL, 2- FVA, 3- CVA.

6.      It’s better to add the section of characterization in the methods and discuss all the equipment used for the characterization with details rather than in the literature of results.

Author Response

In the review of the manuscript titled: Effect of processed volcanic ash as active mineral addition for cement manufacture, overall description is good and the manuscript is written very well. I would like to see this article publish but after some minor modification as follow,

  1. Overall, manuscript must be revised throughout as superscripts and sub-scripts must be checked like;
  1. Page 2, line 77, “CO2”.
  2. Page 4, line 122, “g/cm3”
  • Figure 3, within the figure formula of multilite
  1. Page 5, line 140 and 141, “SiO2”
  2. Page 6 line 170, 171
  3. Figure 4, Inside the “Figure formula for SiO2”,
  • Page 15, second paragraph of conclusion. etc.,

Thank you, it has been corrected

  1. In the introduction portion please cite some references for literature review of your material upto date.

Thank you for your comment. Updated papers from the year 2022 have been entered into the manuscript. The studies referred to are listed below.

  • Khan, K., Amin, M. N., Usman, M., Imran, M., Al-Faiad, M. A., & Shalabi, F. I. (2022). Effect of Fineness and Heat Treatment on the Pozzolanic Activity of Natural Volcanic Ash for Its Utilization as Supplementary Cementitious Materials. Crystals, 12(2), 302.
  • Shahjalal, M., Rahman, J., Haque, A. F., Habib, L., Jalal, K. B., & Rahman, M. M. (2022). Effect of Partial Replacement of Cement with Volcanic Ash on Mechanical Behaviour of Mortar. In Proceedings of the 5th International Conference on Sustainable Civil Engineering Structures and Construction Materials (pp. 33-44). Springer, Singapore.
  • Churata, R., Almirón, J., Vargas, M., Tupayachy-Quispe, D., Torres-Almirón, J., Ortiz-Valdivia, Y., & Velasco, F. (2022). Study of Geopolymer Composites Based on Volcanic Ash, Fly Ash, Pozzolan, Metakaolin and Mining Tailing. Buildings, 12(8), 1118.
  • ABDULMALEK, N., & Chakkamalayath, J. (2022). Cost-Benefit Analysis of Vibrated Cement Concrete and Self-Compacting Concrete Containing Recycled Aggregates and Natural Pozzolana. Journal of Engineering Research.
  • Cultrone, G. (2022). The use of Mount Etna volcanic ash in the production of bricks with good physical-mechanical performance: Converting a problematic waste product into a resource for the construction industry. Ceramics International, 48(4), 5724-5736.
  • Alqarni, A. S. (2022). A comprehensive review on properties of sustainable concrete using volcanic pumice powder ash as a supplementary cementitious material. Construction and Building Materials, 323, 126533.
  1. In the section of method please add the specifications of the XRD machine with company and model.

The description of the model has been introduced in the text.

  1. In my suggestion, it’s better to prepare the origin diagrams (Figure 3, 4 and 5) more attractive by adding the Y axis of intensity (a.u.) as well.

Thank you for your comment. The authors have evaluated the possibility of introducing the original diagrams, but we have considered that the current figures show more clearly the structure of the analyzed ashes and lava.

  1. The authors are requested to add the SEM images with their EDS of all three samples of 1- VL, 2- FVA, 3- CVA.

Thank you for your comment. The morphology of the volcanic material has been determined with scanning electron microscopy (SEM), complemented with EDX to complete the compositional studies. A Hitachi S4800 electron microscope (Tokyo, Japan) was used to study the morphology. For energy dispersive spectroscopy (EDX) determination of the chemical composition of the samples, a Bruker Nano XFlash 5030 silicon drift detector was used.

The results obtained are shown in Figure 6 and Table 4 of the manuscript.

  1. It’s better to add the section of characterization in the methods and discuss all the equipment used for the characterization with details rather than in the literature of results.

Thank you for your comment. The sections have been restructured for a better understanding of the reader.

After complete the revision process, we hope that the revised manuscript does now fully meet the criteria and conditions for publication in Journal of Materials. Thank you very much for your efforts concerning our manuscript. 

Yours sincerely,

Ph D. Francisco Agrela

University of Córdoba

Reviewer 4 Report

The manuscript Effect of processed volcanic ash as active mineral addition for cement manufacture aims to comprehensively utilize volcanic ash and other wastes after volcanic eruption. The manuscript has certain practical significance. The test method is basically correct, and the data is really reliable, but there are the following suggestions for the author's reference.

1. There are a lot of abbreviations in the manuscript, please check their correctness carefully. For example, in Table 5, please explain whether the mix proportion of 10% silica fume in the column SF is correct.

2. The manuscript is more like a complete experimental report. Lack of in-depth theoretical analysis of experimental data. It is suggested that some microscopic tests can be added, such as scanning electron microscopy, X-ray diffraction, infrared spectroscopy, etc. at present, there is little mechanism analysis, which is based on some conjectures of previous literature. It is suggested to prove the hydration mechanism of these volcanic ashes and whether some C-S-H and c-a-s-h gel are generated.

3. The figures and tables in the manuscript need to be strengthened, and the figures and tables need to be more intuitive.

4. It is suggested to add some deficiencies of existing research in the introduction to highlight the significance of this study.

5. Most of the manuscripts are descriptions of experimental data, and it is suggested to add more explanations of deep-seated reasons.

Author Response

Reviewer #4:

The manuscript ‘Effect of processed volcanic ash as active mineral addition for cement manufacture’ aims to comprehensively utilize volcanic ash and other wastes after volcanic eruption. The manuscript has certain practical significance. The test method is basically correct, and the data is really reliable, but there are the following suggestions for the author's reference.

  1. There are a lot of abbreviations in the manuscript, please check their correctness carefully. For example, in Table 5, please explain whether the mix proportion of 10% silica fume in the column SF is correct.

Thank you for your comment, the nomenclatures have been revised.

Regarding the 10% addition of silica fume shown in table 5, yes it is correct, because the regulations UNE EN 197-1 require a maximum addition of 10% for the manufacture of cements. For this reason, the percentage is lower than the rest of the materials studied.

  1. The manuscript is more like a complete experimental report. Lack of in-depth theoretical analysis of experimental data. It is suggested that some microscopic tests can be added, such as scanning electron microscopy, X-ray diffraction, infrared spectroscopy, etc. at present, there is little mechanism analysis, which is based on some conjectures of previous literature. It is suggested to prove the hydration mechanism of these volcanic ashes and whether some C-S-H and c-a-s-h gel are generated.

Thank you for your comment. A microscopy and XRD study of the volcanic material has been added showing the presence of Si, Ca and Al as the basis for C-S-H and C-A-S-H formation.

This work has focused on a previous study to evaluate the application of volcanic ash from the volcano of La Palma as a mineral addition in the manufacture of Portland cement (OPC). For this purpose, a study has been carried out on the physical-chemical properties of the materials, mechanical behavior and pozzolanicity study, comparing it with other commonly used additions. These aspects are the main ones required by the OPC regulations.

Once the feasibility of the application of the volcanic material has been demonstrated, we are currently working on the drafting of an extensive and detailed work focused on the microstructure and on the changes produced in the hydration process, so this information will be included in the future work.

  1. The figures and tables in the manuscript need to be strengthened, and the figures and tables need to be more intuitive.

Thank you for your comment, more figures and tables have been introduced and some have been modified.

  1. It is suggested to add some deficiencies of existing research in the introduction to highlight the significance of this study.

Thank you for your comment, the introduction has been modified through an exhaustive study of recent papers. The importance of this study has been emphasized and the results obtained have been shown in broad outline.

  1. Most of the manuscripts are descriptions of experimental data, and it is suggested to add more explanations of deep-seated reasons.

Thank you for your comment, a revision of the introduction and discussion of results has been made. The analysis of the results obtained has been expanded.

After complete the revision process, we hope that the revised manuscript does now fully meet the criteria and conditions for publication in Journal of Materials. Thank you very much for your efforts concerning our manuscript. 

Yours sincerely,

Ph D. Francisco Agrela

University of Córdoba

Reviewer 5 Report

The English of the manuscript is poor, and must be further improved by a native English speaker who is experienced in this area. 

The current structure of Introduction is not suitable. Small paragraphs with not useful information should be removed or improved. Much useful information can be added. See

https://www.sciencedirect.com/science/article/pii/S1674775521001530

There are many long sentences (for example, last paragraph of the Introduction) that need to be carefully revised.

The novelty of the research is still not cleared. Mechanical activation has been used in several studies for such materials, so what is the difference between this research with the literature? Please explain in the article. 

Figures 2, 6, 7, 8 and 9 and their legends must be redrawn to make the curves/columns easily identifiable for the readers when it is printed in Black&White. 

What is the reason for adopting such mix proportions in Table 5? 

Which started was followed to determine the water absorption rate, density, etc. of the raw materials?

In general, most parts of the manuscript can be further improved (with much emphasis on English writing) before it accepted for publication.

Author Response

Reviewer #5:

  1. The English of the manuscript is poor, and must be further improved by a native English speaker who is experienced in this area. 

Thank you very much, the English of the manuscript has been revised for the second time by a native reviewer. This revision has improved the text.

  1. The current structure of Introduction is not suitable. Small paragraphs with not useful information should be removed or improved. Much useful information can be added. See https://www.sciencedirect.com/science/article/pii/S1674775521001530

There are many long sentences (for example, last paragraph of the Introduction) that need to be carefully revised.

Thank you for your comment, the introduction has been modified through an exhaustive study of recent papers. The importance of this study has been emphasized and the results obtained have been shown in broad outline. The paragraphs have been simplified for better understanding.

  1. The novelty of the research is still not cleared. Mechanical activation has been used in several studies for such materials, so what is the difference between this research with the literature? Please explain in the article. 

Thank you for your comment. A description of the novelty of the work has been included. Volcanic ashes have different properties depending on their origin, so a study is always considered necessary for their possible application. In lines 85-90 the novelty of this study has been explained.

  1. Figures 2, 6, 7, 8 and 9 and their legends must be redrawn to make the curves/columns easily identifiable for the readers when it is printed in Black&White. 

Thank you for your comment, the figures have been replaced.

  1. What is the reason for adopting such mix proportions in Table 5? 

The established dosages have been developed in accordance with the standard UNE EN 197-1.

Regarding the 10% addition of silica fume shown in table 5, yes it is correct, because the regulations UNE EN 197-1 require a maximum addition of 10% for the manufacture of cements. For this reason, the percentage is lower than the rest of the materials studied.

  1. Which started was followed to determine the water absorption rate, density, etc. of the raw materials?

The standard used for the determination of each of the parameters is defined in Table 2.

In general, most parts of the manuscript can be further improved (with much emphasis on English writing) before it accepted for publication.

Round 2

Reviewer 4 Report

After revision, I think the manuscript has been substantially improved.